# Response to comment on "Magnetosensitive neurons mediate geomagnetic orientation in *Caenorhabditis elegans*"

Andres Vidal-Gadea[1]*, Chance Bainbridge[1], Ben Clites[2], Bridgitte E Palacios[2,3], Layla Bakhtiari[3], Vernita Gordon[3], Jonathan Pierce-Shimomura[2]*

[1]School of Biological Sciences, Illinois State University, Normal, United States; [2]Department of Physics, University of Texas at Austin, Austin, United States; [3]Department of Neuroscience, University of Texas at Austin, Austin, United States

**Abstract** Many animals can orient using the earth's magnetic field. In a recent study, we performed three distinct behavioral assays providing evidence that the nematode *Caenorhabditis elegans* orients to earth-strength magnetic fields (*Vidal-Gadea et al., 2015*). A new study by Landler et al. suggests that *C. elegans* does not orient to magnetic fields (*Landler et al., 2018*). They also raise conceptual issues that cast doubt on our study. Here, we explain how they appear to have missed positive results in part by omitting controls and running assays longer than prescribed, so that worms switched their preferred migratory direction within single tests. We also highlight differences in experimental methods and interpretations that may explain our different results and conclusions. Together, these findings provide guidance on how to achieve robust magnetotaxis and reinforce our original finding that *C. elegans* is a suitable model system to study magnetoreception.

DOI: https://doi.org/10.7554/eLife.31414.001

*For correspondence:
avidal@ilstu.edu (AV-G);
jonps@austin.utexas.edu (JP-S)

**Competing interests:** The authors declare that no competing interests exist.

## Introduction

We recently asked whether the nematode *Caenorhabditis elegans* was capable of magnetic orientation (*Vidal-Gadea et al., 2015*). *C. elegans* has proven historically important for the discovery of molecules used to sense odors, mechanical force, osmolarity, and humidity (*Sengupta et al., 1996*; *O'Hagan et al., 2005*; Colbert et al., 1997; *Russell et al., 2014*). Notably, each of these molecules share conserved functions in higher animals (*Tobin and Bargmann, 2004*; *Arnadóttir and Chalfie, 2010*; *Filingeri, 2015*). If *C. elegans* displays magnetoreception, potentially conserved molecular bases for this sensory modality may be studied using similar approaches. Using three distinct behavioral assays, we discovered that this tiny worm could orient its movement to artificial magnets or to the earth's magnetic field (*Vidal-Gadea et al., 2015*).

More recently, *Landler et al., 2018* performed additional sets of behavioral experiments to confirm whether *C. elegans* orients to magnetic fields. They reported negative results for all three experiments and conclude that *C. elegans* is not a suitable model system to study the molecular basis for magnetoreception. On first inspection, the experiments done by *Landler et al., 2018* resemble those from our study with additional levels of control. However, there are important differences in their experimental methods, controls, and execution which might have contributed to their negative results.

*Landler et al., 2018* also raised conceptual issues with our findings and interpretations. First, they suggest that *C. elegans* should not be able to orient to the strong magnetic field used in our

magnetotaxis assay. Second, they suggest that a tentative explanatory hypothesis that we put forward– that *C. elegans* strains isolated from different locations on the globe may migrate at a specific angle to the magnetic field, perhaps as a way to orient optimally up or downwards when burrowing is infeasible. We address the first issue by showing that the magnetic field provides directional information in our magnetotaxis assay, allowing us to predict the unusual tracks they made in our original 2015 study. We finish by identifying plausible mechanisms for how worms may use the directional information provided by a magnetic field to migrate along a specific vector.

## Results and discussion

### Overt differences in experimental methods

Landler et al. attempted to reproduce our results with British worms by performing modified versions of three of our experiments. These modifications included worthwhile control measures and analysis that differed slightly from our original study. Unfortunately, it appears many of these experiments deviated from our described methods. For each experiment, they found negative results concluding that *C. elegans* may not orient to magnetic fields. Below, we discuss differences in experimental methods, analysis, and interpretation that may explain their failure to match our results.

#### Animal satiation states

One of the major differences between our methods and theirs was in the duration of the assays. In *Vidal-Gadea et al., 2015* we experimentally determined and reported that 30 min away from food was sufficient to flip the magnetotaxis behavior of the worms from positive to negative. This was initially unexpected, because *C. elegans* does not flip its orientation preference after 30 min away from food for other orientation behaviors including chemotaxis to benzaldehyde. We therefore described this time as sufficient to induce the 'starved' state in animals, and went on to perform several experiments with worms in the 'fed' or 'starved' states. For 'starved' assays, we ensured that worms were away from food for 30 min prior to starting an experiment. From reading *Landler et al., 2018* it is now clear that we did not explicitly mention that this definition of starvation implied that for worms to be tested in the 'fed' state, animals would need to complete their assay within 30 min.

With practice, we found that we could run our behavioral assays to near completion within this 30 min time window and did not need to immediately tally immobilized animals. This allowed us to run many assays in parallel without having to stop to conduct the time-consuming tallying step for each pipette or plate before starting the next. Tallying animals in the magnetotaxis assay, however, was a much simpler (and faster) procedure, which we could do at the 30 min mark. Therefore, it is important to note that we ran assays so that worms migrated to a particular direction within 30 min of initial removal from seeded growth plate.

It is clear from *Landler et al., 2018* that they decided to modify our magnet assay to last 60 min rather than 30 min precisely because they continued to see moving animals all the way until this time point (see their Methods). Unfortunately, this also implies that many worms participating in the assay (which they described was a sufficiently large number to make them deviate from our protocol) would have transitioned to the 'starved' state. By our described definition of 'fed' and 'starved' (also adopted by Landler et al.) they report testing animals under both 'fed' (first half of the assay), and 'starved' (second half of the assay) conditions. This issue may have been obviated when Landler et al. tested pre-starved worms (their Figure 3B); however, no-magnet controls and horizontal-oriented tube controls for these assays were not reported (see below). We believe this singular, and crucial, difference might explain their different results.

#### Animal rearing

A second major difference between our methods involved the rearing conditions for our animals. Like many *C. elegans* labs, our worms were grown in laboratories maintained at 20°C. We deliberately kept animals away from artificial magnetic fields (generated by electrical equipment or wiring).

Landler et al. grew their animals in incubators. This attempt at controlled culturing could have accidentally grown their animals under extreme magnetic and electric fields conditions generated by their incubators. This is not trivial. Recent studies demonstrated that extreme magnetic inhomogeneities are produced within these type of devices (*Makinistian and Belyaev, 2018*). For example, animals cultured mere centimeters apart would be exposed to fields differing by up to a factor of 36. This includes hypomagnetic field areas, where the absence of magnetic fields might affect the development of magnetic organs. Therefore, it appears that while rearing animals at 20°C Landler et al. might have unintentionally cultured animals under wildly variable magnetic conditions. This method might also produce preferences for a cultivation temperature in animals that could confound behavior when they are tested in a chamber with a different ambient temperature. We are not certain of what effect these conditions might produce in the magnetic machinery or behavior of worms, however we think it prudent to control magnetic exposure of animals to be used in magnetic studies.

## Additional differences in experimental methods

We next describe additional differences between our experiments that might have further contributed to their observations.

### Burrowing assay

*Landler et al., 2018* assayed whether N2 worms injected into agar-filled cylinders burrowed up or down. They report no bias for burrowing up or down, with or without an imposed inverted magnetic field.

The magnetic coil systems used in the two studies differ. In our 2015 study, we used three sets of four-coil systems described by Merritt (*Merritt et al., 1983*) which was reported to produce more stable test fields for larger samples (e.g. 50 vs 20 cm$^3$) than with the Helmholtz loops (*Kirschvink, 1992*; *Magdaleno-Adame et al., 2010*) used by *Landler et al., 2018*. Before we conducted our burrowing assays inside our magnetic coil system, we measured the field properties throughout the test volume to ensure that our system produced a homogeneous field. Landler et al. (2017) did not report such measurements of homogeneity.

*Landler et al., 2018* suggested that unintended temperature gradients generated by our coil system may have resulted in our reported observations. However, aware of this possibility, we used high sensitivity thermometers (0.01°C) to quantify temperature changes (in both magnet, and magnetic coil system assays). A two-way ANOVA (N = 5, p=0.123) showed no significant difference in temperature between cancelled field and one-earth field (Figure 2—figure supplement 1 in *Vidal-Gadea et al., 2015*). As an extra precaution, we regularly rotated the orientation of our magnetic coil system to a random position before each assay, and used a small fan to circulate air through the coil to prevent temperature gradients. While clearly understanding the risk of unintended thermal gradients, *Landler et al., 2018* do not describe doing any such controls and they do not present associated temperature measurements. However, from their discussion they seem to believe that the use of double wrapped coils prevents the generation of meaningful temperature gradients. This is incorrect; all current passing through a metal conductor generates heat. Therefore, under both their test and control conditions, heat would be generated by current passing through their system, potentially building up within the small enclosed metal room used to shield worms from foreign magnetic fields. We welcome their additional control of a mu-metal shielded room as an improvement to minimize potential magnetic contamination, but caution that environmental parameters should be controlled further by positioning the coil system at random orientations within the room, and by empirically measuring and mitigating the thermal gradients that must necessarily develop as current powers any coil system as we did in our original study.

Our burrowing experiments tested in the natural earth field showed that worms migrated differentially based on their global site of origin and satiation state. This observation undermines the likelihood of temperature gradients, or magnetite contamination, being responsible for our results. Furthermore, our findings that worms lacking the transduction channel encoded by the *tax-4* gene, or by worms with genetically ablated AFD neurons, failed to burrow preferentially up or down in earth's natural field strongly point to the involvement of these neurons and molecules in this behavior (*Vidal-Gadea et al., 2015*). None of these results are mentioned in *Landler et al., 2018*.

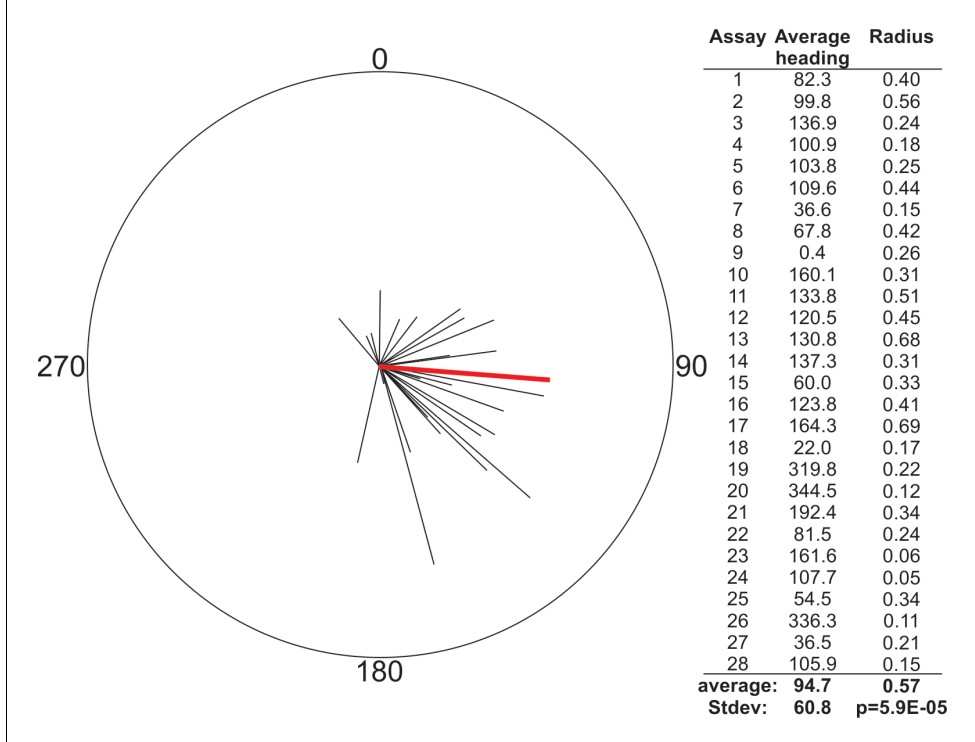

**Figure 1.** Reanalysis of population data from horizontal field assay in *Vidal-Gadea et al., 2015* confirms strong orientation with respect to imposed magnetic field. Well-fed N2 worms were placed in the center of a 10 cm diameter plate and allowed to migrate freely for 30 min as an earth-strength magnetic field was imposed across the surface of the plate. Worms were trapped by sodium azide at the perimeter. The vector averages for each of the 28 assays (*black lines*) are plotted as well as the average of these vectors (*red line*). Vector average values are listed on right. This analysis found a significantly biased vector average of 94.7° (p<0.0001) that was not statistically different from our previously reported value of 132° based on the analysis of individual worms.

DOI: https://doi.org/10.7554/eLife.31414.002

## Horizontal plate assay

*Landler et al., 2018* tested how worms migrate to the edge of a 10 cm diameter plate in a horizontal magnetic field where they were trapped by azide at the edge. Unlike our study, they found no significant degree of orientation in their migration.

In *Landler et al., 2018*, they point out that we treated each worm as an individual in our horizontal field assays when performing statistical analysis, which could cause a type one error if worms did not act independently. While we believe that the individual timing and trajectory of each worm makes them independent, which justified our choice of statistical analysis, we nevertheless re-analyzed our data averaging the mean heading of worms in each assay as they did. Similar to our previous report, we found that in all but one out of 28 assays, worms displayed a significant migratory preference (*Figure 1*). The average heading of the assays changed somewhat from the result reported in *Vidal-Gadea et al., 2015*, but not significantly so.

*Landler et al., 2018* also noted that all worms in their magnetic field conditions are set by an experimenter not involved in the analysis. This blinding protocol was also the case for our original study, but not explicitly mentioned. Anecdotally, we expected British worms to migrate towards magnetic north, just like magnetotactic bacteria, but remained puzzled for months when our results unexpectedly showed them consistently migrating at a 132° angle with respect to magnetic north. This illustrates how our expectations did not affect our analysis or results. Indeed, we still do not know why worms prefer this particular angle, although we presented a parsimonious explanation that worms may choose a migratory direction based on the inclination of their native field.

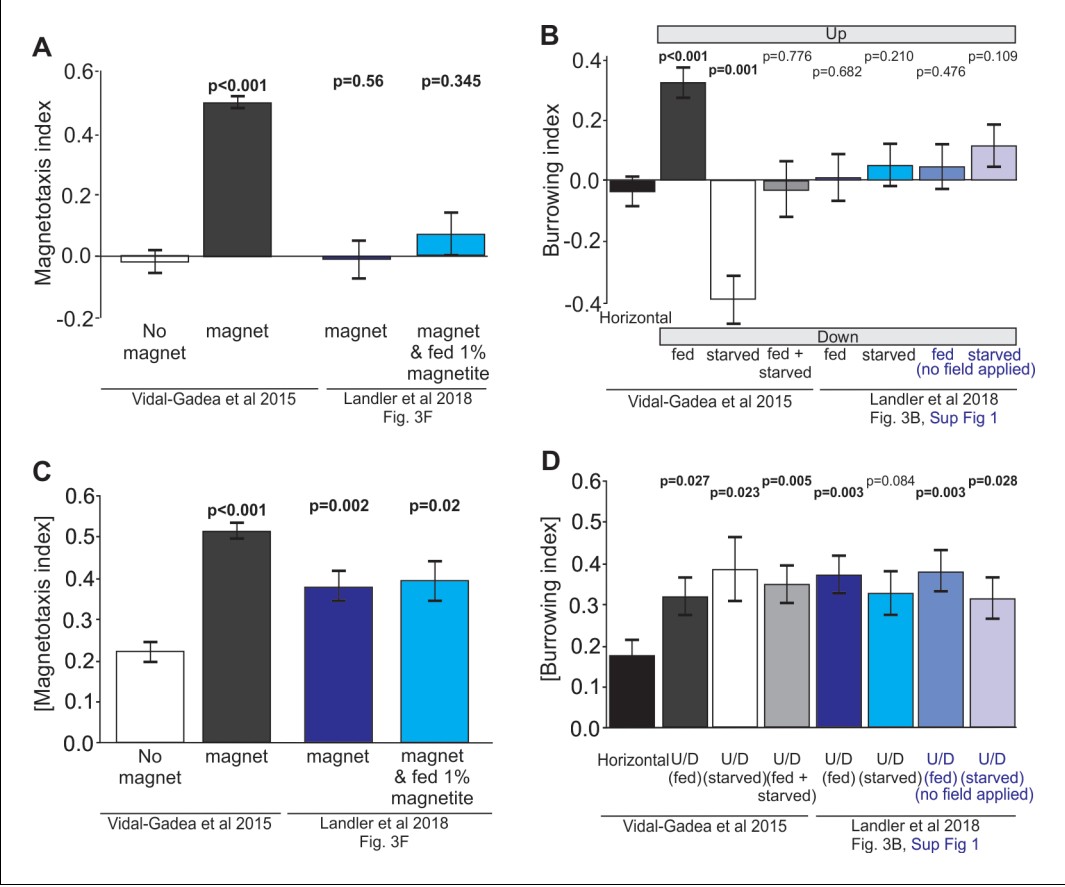

**Figure 2.** Reanalysis of data suggests that positive results may have been masked in *Landler et al., 2018* by testing worms in both fed and starved states and omitting controls. (**A**) Comparison of magnetotaxis data reported by *Vidal-Gadea et al., 2015* and *Landler et al., 2018* obtained by measuring their plots. Figure 3B and F, and Figure 3B, Figure 3—figure supplement 1 were reproduced from *Landler et al., 2018*; published under the terms of the Creative Commons Attribution license (http://creativecommons.org/licenses/by/4.0/)). Because Landler et al. omitted no-magnet control data, we used no-magnet control data from *Vidal-Gadea et al., 2015*. We found that *Landler et al., 2018* worms fed OP50 bacteria (or OP50 plus 1% magnetite) show no significant orientation versus no-magnet control worms. (**B**) Under the hypothesis that Landler et al might have combined fed and starved worms because their assays were run for twice as long, we used the absolute value of the magnetotaxis index to reveal evidence that worms display a biased migration in the presence of a magnetic field (irrespective of the towards or away sign of their migration). We found that both magnet treatments in *Landler et al., 2018* resulted in significantly biased migration when compared with no-magnet controls. (**C**) We also analyzed burrowing data from *Landler et al., 2018* and used our horizontal controls because they were omitted in *Landler et al., 2018*. We demonstrate that combining data from fed and starved worm abolished significant burrowing indexes that were otherwise observed from each of these populations. Similarly, comparison of *Landler et al., 2018* burrowing indices to our horizontal controls (N = 24) revealed no burrowing bias in their field up results for either fed or starved conditions. (**D**) However, when we compared the absolute value of burrowing bias we found that our combined fed + starved group, as well as Landler et al.'s 'fed' worms now showed significant bias when compared to horizontal controls. All tests based on Mann-Whitney Ranked Sum Tests.

DOI: https://doi.org/10.7554/eLife.31414.003

## Magnetotaxis assay

Landler et al. also performed a modified version of our magnetotaxis assay, reporting that worms exhibited no preference for the magnet when compared with a Wilcoxon signed rank test to a control group (although the control group is not described or plotted in their manuscript). However,

Landler et al. chose to extend their assay time. We feel it is unacceptable that they fail to display their control data due to implications described below.

In their manuscript, *Landler et al., 2018* offered no actual replication for our magnetotaxis experiments, opting instead for modified protocols. For example, Landler et al. chose to extend their assay time to one hour to include worms that may have switched to a starved state. As described above, it is therefore likely that by testing fed and starved animals alongside one another, they effectively combined and measured positive and negative magnetotaxis.

Landler et al. conducted an additional control in the form of a magnetic assay with worms fed 1% magnetite throughout their cultivation. They reported a barely significant (p<0.04) improvement over chance performance data that they chose not to plot; however, their assay was still not significantly different from animals not fed iron that migrated in the presence of a magnet. Their observation that worms contaminated with magnetite fail to migrate to a magnet as readily as in our assays, coupled with the lack of significant difference between their 'contaminated' vs 'uncontaminated' samples provides strong evidence that contamination with magnetite is likely not responsible for our observations. However, we take this conclusion with caution since their one-hour assay might have masked potentially significant effects arising from their magnetite enrichment.

Magnetite is pervasive in soils adjacent to the decaying fruit and plant matter where worms live (*Stanjek et al., 1994*; *Moskowitz, 1995*; *Schulenburg and Félix, 2017*; *Schulenburg and Félix, 2017*). It is likely that worms in their natural habitat may have access to much greater amounts of iron than those provided for them in the lab via a diet of *E. coli* on an agar surface. If *C. elegans* does use magnetite to build its own magnetic field detector, it is possible that enrichment of the lab culture conditions may result in increased magnetic indices for adults, or even succeed in enabling larval stage worms to perform this behavior better than what we previously reported (*Bainbridge et al., 2016*). While we consider this experiment informative, we do not agree with *Landler et al., 2018* in their suggestion that ingestion of magnetic particles might confer the ability to migrate within magnetic fields to *C. elegans*. This would not explain how worms migrate at different angles in a satiety-dependent manner, or that this behavior can be systematically abolished and rescued by cellular and molecular tinkering. To our knowledge, ingestion of magnetite or iron is yet to be demonstrated in any animal to be sufficient to confer the ability to migrate towards and/or away from magnets using their own power. Thus, although exogenous iron has been suggested to contaminate cells proposed to be magnetoreceptors (e.g. *Edelman et al., 2015*), in contrast to *Landler et al., 2018* claim, there is no example of false positives associated with actual magnetoreceptive behavior or physiology displayed by any wild-type animal. Additionally, in our initial study, we tested more than 30 mutants with impairments in a broad range known sensory modalities that *C. elegans* has been shown to detect (including temperature, electric fields, soluble and volatile chemicals, touch, light, osmolality, acidity, water, $CO_2$, and $O_2$). Only mutants related to AFD function were unable to orient to magnetic fields (*Supplementary file 1*).

One important control provided in our manuscript but left out in *Landler et al., 2018* was the response of worms in the absence of a magnet. In their Figure 3F the authors display results for two assays in which worms are in the presence of a magnet, with the only difference being whether or not their diet was enriched with magnetite. No control assays where worms tested in the absence of a magnet are provided. Additionally, worms tested with a magnet are confusingly labelled 'baseline'. We find the omission of this no-magnet control, and the nomenclature used, concerning. Based on our experience, fed worms display magnetotaxis indices with high positive values, and starved worms display indices with high negative values. If Landler et al. inadvertently combined in effect starved and fed worms in their assays, then we expect them to observe a broad range of magnetotaxis indices with an average centered at zero. These results would starkly differ from control assays with no magnet, where indices would more tightly cluster around zero (*Vidal-Gadea et al., 2015*). If their no-magnet control data are significantly more narrowly distributed around zero than their test data, this would suggest that the broad distribution of indices obtained in their magnet assays is likely the result of combining fed and starved animals that are in fact orienting to the magnetic field, but in opposite directions thus resulting in an average calculated index of zero.

To investigate the possibility that Landler et al. had obtained positive results that were masked by testing worms in both fed and starved states, we reanalyzed their data. First, we plotted their original results from Figure 3F in bar format alongside results from our original study including our no-magnet control (*Figure 2A*). Next, to control for satiation dependent reversals in magnetotaxis

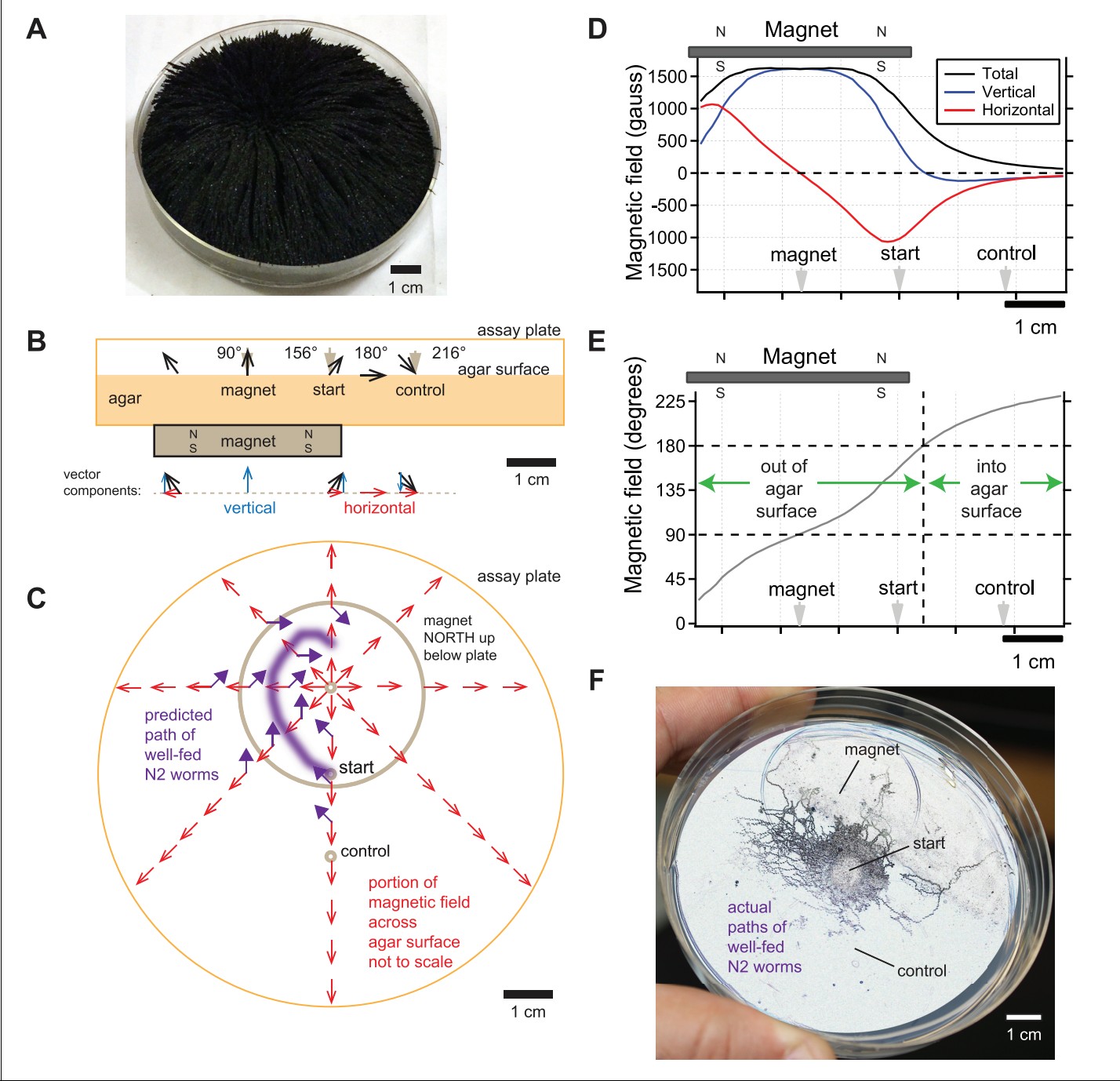

**Figure 3.** Directional information in magnetic field predicts *C. elegans* magnetotaxis trajectory. (**A**) The direction of iron filings scattered across an assay plate reveals the general shape of magnetic field emanating from a 1.5' diameter magnet, north-facing up beneath the plate. (**B**) Side view of magnetic field lines and their vertical and horizontal components across the surface of the agar-filled plate. Magnet and plate shown to scale. Field line strength not to scale. Gray arrowheads denote start location for worms and points where azide was spotted above the magnet and control goals. (**C**) Top view of horizontal component of magnetic field (red arrows) across the surface of the agar-filled plate. Note that magnetic north points directly away from the center of the magnet everywhere on the plate. Wild-type N2 worms prefer to move at 132° to magnetic north, which predicts the trajectory (purple arrows and line). Field lines not to scale. (**D**) Strength of the total magnetic field and its vertical and horizontal components across the agar surface. (**E**) Inclination angle of the magnetic field across the agar surface. (**F**) Majority of observed trajectories for N2 worms in the magnetotaxis assay arc left of magnet consistent with prediction.

DOI: https://doi.org/10.7554/eLife.31414.004

preferences, we plotted the absolute value of magnetotaxis indices (*Figure 2B*). With this alternative analysis, we found that worms migrating in the presence of a magnet displayed a significant migratory bias when compared to worms migrating in the absence of a magnet (*Figure 2B*). *Landler et al., 2018* reported that their magnetite-enriched worms did not migrate significantly better than worms that were not fed magnetite (p=0.652). Our reanalysis confirms their finding, but also shows that their worms in both test conditions displayed a significant difference from our no-magnet control. We also reanalyzed burrowing data plotted in Figure 3B and Figure 3—figure supplement 1 from *Landler et al., 2018*. Once again, we provided missing controls in the form of worms burrowing in horizontally oriented pipettes and compared these to data from both groups (*Figure 2C&D*). We found an identical pattern of results where the absolute value of the burrowing index values for all six test group worms were higher than our horizontal control worms with three of the four groups showing significant differences (*Figure 2D*, Mann-Whitney Rank Sum Test). Importantly, our reanalysis suggests that had no-magnet, or horizontal controls been included and analyzed by Landler et al., they would have obtained positive results for magnetic orientation in *C. elegans*.

## New behavioral experiments support original study

### Reproduction by independent labs

Since our initial description of this behavior in *C. elegans* (*Vidal-Gadea and Pierce-Shimomura, 2012a*), we are aware of several groups joining the study of magnetic field detection using nematodes. While conducting our original study, Ilan et al. (2013) reported that parasitic nematodes migrated preferentially south over north when placed in a magnetic field. We recently became aware of a group at the University of Quilmes, Argentina who independently reproduced our findings with minor modifications (*Vidal-Gadea et al., 2018*).

*Landler et al., 2018* notes that a study by *Njus et al., 2015* reported that worms failed to respond to magnetic fields. Njus et al. restricted their study to crawling velocity and omega bends and not orientation. Nevertheless, in Figure 6 of *Njus et al., 2015*, they show a 10 ± 7% to 90 ± 20% change omega bends when a 5-mT magnetic field was introduced or removed respectively. Such difference in turns would have a significant effect on course trajectory and orientation. Rather than comparing these paired measurements to each other, Njus et al. compared them to the number of worms turning in the absence of a magnetic stimuli for which they report an average of 70 ± 50%. A 50% variability in omega bends is surprising and not consistent with previous reports in the literature (e.g. *Vidal-Gadea et al., 2012*), or even with the variability they report for the rest of their data (22.5 ± 8%, obtained by measuring and averaging standard deviations from test conditions reported by *Njus et al., 2015*: Figure 6). Not surprisingly, no test condition was significantly different from such a variable control. Therefore, the *Njus et al., 2015* study appears to offer little evidence to counter the idea that *C. elegans* orients to magnetic fields.

## Conceptual issues regarding magnetic orientation in *C. elegans*

In addition to methodological issues, *Landler et al., 2018* raise two conceptual issues regarding our original study that we address below.

### Magnetotaxis assay trajectories

How do worms move in our magnetotaxis assay? As described above, worms are placed in the center of an agar-filled Petri plate with a 0.29 T strength, 1.5-inch diameter, neodymium magnet placed north-side facing up 1 cm beneath the agar surface on one side of the plate (*Figure 3A–C*). Azide is pipetted on magnet and control sides to immobilize worms that reach either location. Landler et al., (2017) correctly point out that the intensity of the magnetic field generated by the magnet is many times stronger than the earth's field. They ask how could worms orient to this magnetic gradient if they never encountered magnetic fields this size during their course of evolution.

We agree with Landler that the worms are unlikely to distinguish the difference between magnetic field intensities higher than earth strength. This is consistent with our finding that the AFD magnetosensory neurons failed to generate larger responses when presented with larger than earth-strength fields (*Vidal-Gadea et al., 2015*). Instead, we believe that *C. elegans* pays attention to the directional information contained in the magnetic field gradient. To help visualize the magnetic field

in our magnetotaxis assay, we scattered iron filings across the agar surface of the assay plate (*Figure 3A*). The filings stand straight up directly above the center of the magnet, indicating that the magnetic field is perpendicular to the agar surface here. Filings tilt at increasingly shallow angles as the distance from the magnet center increases until they become parallel with the plate surface. This simple experiment shows that magnetic field lines pierce the surface of the agar at a variety of angles in a radial pattern which could provide abundant directional information pointing away from the center of the magnet in our, and Landler et al.'s, experiments.

We next calculated the direction and magnitude of the magnetic field across the assay plate. This was done using the online calculator by K and J Magnetics, and validated experimentally with the aid of a magnetometer. We found that the magnetic field was strongest at the center of the magnet and began to strongly dissipate near the inner edge of magnet (*Figure 3D*). The horizontal component of the field that is parallel to the plate surface was zero at the magnet center and increased away from this point up until near the inner edge of the magnet (red line, *Figure 3D*). The sign of the horizontal component switched at the center of the magnet reflecting how the field lines point radially away from the magnet center. The angle of field penetration varied across the assay plate as expected from the iron filings (*Figure 3A,B&E*). The field pierces out of the agar surface at 90° only at the center of the magnet, and starts tilting until it reaches 180° about 21.5 mm from the center of the magnet (*Figure 3B&E*). Beyond this point, the magnetic field starts to tilt further, piercing into the surface of the agar. Therefore, the magnetic field in the magnetotaxis assay varies in polarity and inclination which we hypothesize worms may use as a cue to orient (*Figure 3B&C*).

Armed with an empirically validated model of the magnetic field in our plates, we can predict how worms move in the magnetotaxis assay. We previously found that in a uniform horizontal field, N2 worms preferred to migrate approximately 132° away from magnetic north when well fed (*Vidal-Gadea et al., 2015*). Given this preference, we expect that worms would make a leftward arc when viewing the assay plate from above (*Figure 3C*). This is because worms started at the center of the assay plate would consistently bear 132° away from the horizontal field lines (purple arrows, *Figure 3C*). To test this prediction, we retrieved photographs of assay plates from our original 2015 study. We found that a significant portion of well-fed worms migrate towards the magnet along the left side of the plates (*Figure 3F*). This unusually asymmetric arced trajectory contrasts greatly from the typical symmetric trajectory that worms make when migrating to the peak of an attractant chemical gradient during chemotaxis (e.g. *Pierce-Shimomura et al., 1999*).

Taken together, this reanalysis of data from our 2015 study unifies the migratory patterns observed in all three behavioral assays and yields new predictions strengthening our original findings.

## Magnetic orientation in three-dimensions

In our original study, we observed that different wild *C. elegans* strains isolated from different locations on the earth migrate at a different particular angle relative to magnetic north. For instance, relative to magnetic north, well-fed worms from Britain accumulate on average at 132°, Australian worms at 302°, and Hawaiian worms at 121°. Moreover, when worms were starved, each strain migrated ~180° relative to the preferred angle when well fed. In our study, we made the parsimonious hypothesis that these different angles may relate to the different inclination angle of the earth's magnetic field at each location.

Given these results, *Landler et al., 2018*, and originally *Parthasarathy, 2015*, suggested that if worms simply migrated at a fixed angle relative to the magnetic field, then a population of worms dispersing from a single point outward would form a cone-shaped trajectory when burrowing in three-dimensional space. The apex angle of the cone would be twice the preferred angle and only one line along the cone would aim correctly up or down.

We were also puzzled how worms migrated at a particular angle to 2D magnetic field in our horizontal plate assay. With our new analysis above, however, this unexpected behavior appears to be consistent across all three of our magnetic orientation assays. It is important to clarify that these results provide evidence demonstrating that worms do not simply migrate at a fixed angle with respect to magnetic north. If worms simply migrated at a fixed angle to the field in a conical trajectory, then worms would have accumulated at two positions symmetrical about magnetic north on the edge of the horizontal plate. Instead, we found in that worms consistently accumulated at only one position. We observed this result in six out of six conditions – three independent wild-type

strains in both fed and starved states. Likewise, if worms simply migrated at a fixed angle with respect to magnetic north in the magnetotaxis assay, then they would have moved both towards and away from the magnet in similar proportions based on our new analysis above.

We still do not understand how or why worms behave this way, and are not wedded to any particular hypothesis. As the first study of magnetic orientation in *C. elegans*, we do not feel that we must provide a mechanistic explanation for a finding that we do not yet fully understand. We tend to agree with Landler et al. and others in their observation that magnetic orientation alone is unlikely to be the only way worms navigate vertically. In our original study we did not hypothesize that worms orient in three dimensions solely by employing their magnetic sense. Organisms known to use the earth magnetic field in orientation also rely on additional sensory modalities to accomplish their behaviors. This is true of magnetotactic bacteria (which combine magnetotaxis with chemotaxis) and birds (which rely on vision for much of their migrations) (*Chen et al., 2010*; *Muheim et al., 2016*). For *C. elegans*, the AFD sensory neurons are clearly established as thermosensory, but are also involved in humidity, and $CO_2$ detection (*Mori and Ohshima, 1995*; *Bretscher et al., 2011*; *Russell et al., 2014*). As we noted in *Vidal-Gadea et al., 2015*, these environmental parameters are stratified vertically in the soil, although the direction of their gradients can vary independently for each parameter. Consider for example the reversal of the vertical temperature gradient in the soil during daytime *vs* nighttime, or reversals in humidity gradients during a rainfall *vs* following a rain. These complex cues likely provide worms with reliable information about the vertical dimension. However, because each of these cues regularly reverse their gradients, they are less likely to provide reliable orientation information (i.e. which way is up or down). We hypothesize that magnetosensation allows worms to disambiguate directional information associated with other environmental cues.

## Conclusion

Magnetic orientation may be challenging to test in *C. elegans*, but worthwhile to get a foothold in discovering some of the first evidence for cellular and molecular basis for magnetoreception in animals.

# Materials and methods

## Estimation of magnetic field

We used the K and J Magnetics magnetic field calculator to approximate field strength over distance and validated the resulting field components with our DC milligauss meter model mgm magnetometer (Alphalab, Utah).

## Statistics

Vectorial data were analyzed as previously described (*Vidal-Gadea et al., 2015*) using Circular Toolbox for Matlab (Mathworks). Following Landler et al., (2017), animals were not pooled but each assay was rather treated as a unit. We conducted Rayleigh tests to determine probability of deviation from circular distribution. Non- parametric groups were compared using Mann-Whitney Ranked Sum tests.

# Acknowledgements

We wish to acknowledge the *Caenorhabditis* Genetics Center which is supported by the National Institutes of Health, as well as NIH grants to AV-G (R15AR068583) and JP (R01NS075541 and 1RF1AG057355).

# Additional information

### Funding

| Funder | Grant reference number | Author |
| --- | --- | --- |
| National Institutes of Health | R15AR068583 | Andres Vidal-Gadea |
| National Institutes of Health | R01NS075541 | Jonathan Pierce-Shimomura |

| National Institutes of Health | 1RF1AG057355 | Jonathan Pierce-Shimomura |

The funders had no role in study design, data collection and interpretation, or the decision to submit the work for publication.

### Author contributions
Andres Vidal-Gadea, Supervision, Writing—original draft, Writing—review and editing, oversaw project; analyzed data; helped with conceptualization; developed novel methods; conducted experiments; wrote original and final draft; Chance Bainbridge, Writing—original draft, Helped with conceptualization; Developed novel methods; Conducted experiments; edited original draft; Ben Clites, Writing—review and editing, Helped with conceptualization; Developed novel methods; Conducted experiments; Edited original draft; Bridgitte E Palacios, Conducted experiments; Layla Bakhtiari, Vernita Gordon, Helped with conceptualization; Edited original draft; Jonathan Pierce-Shimomura, Software, Supervision, Writing—original draft, Writing—review and editing, Oversaw project; Analyzed data; Helped with conceptualization; Ddeveloped novel methods; Conducted experiments; Wrote original and final draft

### Author ORCIDs
Andres Vidal-Gadea (iD) http://orcid.org/0000-0001-5981-5528
Jonathan Pierce-Shimomura (iD) http://orcid.org/0000-0002-9619-4713

### Decision letter and Author response
Decision letter https://doi.org/10.7554/eLife.31414.009
Author response https://doi.org/10.7554/eLife.31414.010

## Additional files

### Supplementary files
• Supplementary file 1. Combinatorial analysis shows magnetic orientation is distinct from other sensory modalities.
DOI: https://doi.org/10.7554/eLife.31414.005

• Transparent reporting form
DOI: https://doi.org/10.7554/eLife.31414.006

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
