## [Decision Letter]

Thank you for submitting your article "Response to Comment on "Magnetosensitive neurons mediate geomagnetic orientation in *Caenorhabditis elegans*"" for consideration by *eLife*. Your article has been reviewed by three peer reviewers, and the evaluation has been overseen by a Reviewing Editor and Eve Marder as the Senior Editor. The following individuals involved in review of your submission have agreed to reveal their identity: Markus Meister (Reviewer #2); Pavel Nemec (Reviewer #3).

The reviewers have discussed the reviews with one another and the Reviewing Editor has drafted this decision to help you prepare a revised submission.

Summary:

The rebuttal from Vidal-Gadea has three main parts: 1) A specific technical concern regarding Landler's attempt at replication. This is legitimate and Landler could address it with a reasonable bit of additional analysis. 2) A long list of potential reasons why magnetotaxis experiments might fail. This line of argument supports Landler's conclusion that *C. elegans* magnetotaxis is not a robust effect and thus not a suitable model system. 3) A report on new experiments on the subject that appear to confirm magnetotaxis in *C. elegans*. These don't directly address the Landler report, they use somewhat different methods, and should be published separately. Thus, the reviewers and BRE feel that a much reduced manuscript that does not introduce new data is now appropriate, and feel that any entirely new experiments belong in a new, separate paper.

Specific comments:

1) In subsection “Magnetotaxis assay”, the authors state (referring to their 2015 paper) "the cGMP-gated cation channel TAX-4 was necessary and sufficient in AFD neurons for magnetic orientation" While they did show TAX-4 was required, they did not show that is was sufficient (i.e. that expression of TAX-4 in a non-magnetically sensitive cell conferred magnetic sensitivity). In fact, TAX-4 is expressed in many sensory neurons (AWB, URX, ASE, AQR….) that are not seemingly involved in magentosensation. This statement needs to be changed.

2) In subsection “Overt differences in experimental methods”: The main technical concern regarding Landler's experiments is that the movement assays were conducted over a longer period of time (60 min vs 30 min). Vidal-Gadea contends that over this period the worms could have changed from "fed" to "starved", leading them to switch the polarity of magnetotaxis. If so, then Landler may have observed a mix of worms moving in opposite directions. Note that this concern applies to some but not all of the Landler experiments, for example the starved worms in their Figure 3B were presumably always starved, and here Landler's results are in direct conflict with Vidal-Gadea's.

3) In the same subsection: A long list of what look like minor differences between the experimental protocols in the two labs. For none of these differences (electrically shielded pipettes, plastic caps vs parafilm, brass vs aluminum for the sham magnet, etc.) is there a plausible argument how they would affect magnetotaxis. If in fact magnetotaxis is real, but so fragile that it depends in some intricate manner on the assay tube material or on large-scale field homogeneity or on minuscule temperature gradients, then one would have to agree with Landler that this just isn't a robust model system for magnetotaxis.

4) In subsection “Additional differences in experimental methods”, the authors cite the requirement for tax-4 and the AFD neurons for as evidence against magnetotaxis being an artifact of temperature gradients. But as the authors must know, AFD neurons are exquisitely sensitive to very small temperature fluctuations, and their temperature responses (which depend on tax-4) are far more robust than their responses to earth-strength magnetic fields. To this reviewer, the involvement of AFD undermines rather than supports their contention that temperature is not involved in the behavior they observe. This section should probably be cut.

5) In subsection “Horizontal plate assay”, indeed, levels of significance reported in the original paper (Vidal-Gadea et al., 2015) were based on an inappropriate number of degrees of freedom. This is because individual worms were used as replicates although they were tested within the same Petri dish and were therefore not independent of one another. Treating them as such violates a key assumption of the statistical tests used and leads to inflated type 1 error. In the response to Lander et al. Vidal-Gadea et al. claim that reanalysis of the data using mean headings of the worms had no major effect on results. They should either refrain from this argument or present the re-analysed data (preferably in from of a supplementary table).

6) In subsection “Magnetotaxis assay*”*, it is stated that " worms in their natural habitat may have access to much greater amounts of iron than those provided for them in the lab…." We don't know what evidence the authors have for this statement. The current view on *C. elegans* ecology (e.g. Felix and Braendle, 2010) is that it is not a soil-dweller but rather colonizes decaying fruit and plant matter. Please remove this statement.

7) In many places the authors state that "worms migrated differently based on their global site of origin" with the specific implication that their internal compass is set to give a consistent up/down migration direction given the location of the population on the globe. These results are too preliminary to firmly draw this conclusion. Although many wild isolates of *C. elegans* have been described and are available from the stock center, the authors have only tested 3. Moreover, one of these, the N2 lab strain, is not really a good example of a "British" wild strain N2 was grown in liquid culture for many years (in Berkeley) followed by many more years of culture on agar plates before it was frozen. During these thousands of generations of "domestication", it is known to have acquired multiple mutations adapting it to growth in a lab environment. To establish a firm link between the local magnetic field orientation and the preferred migration direction of wild populations, it would be important to sample a wider selection of genuinely wild *C. elegans* isolates, and ideally cross them to get some insight into how magnetic preference segregates genetically. While this is more a critique of the original 2015 paper than anything else, it is not clear that the somewhat preliminary wild isolates data strengthens the authors' case very much.

8) In subsection “Important experimental variables for magnetic orientation in *C. elegans”*: An additional long list of environmental factors that could lead magnetotaxis experiments to fail. "Because of these demanding physiological and environmental parameters that the worms must be in before they will orient to magnetic fields, we hesitate to suggest which parameter may have caused trouble in Landler et al., 2018." This raises some fundamental concerns: The claim that experiments "work" only under a very specific constellation of conditions is a feature of what Langmuir once dubbed "pathological science". This belief opens a large number of degrees of freedom that the experimenter can use for data selection. If one day the magnetotaxis assay "didn't work", the student is tempted to look at the hygrometer and conclude that the weather was either too humid or too dry and exclude that data set from analysis. Perhaps Vidal-Gadea and colleagues took special precautions to guard against such risks of self-deception, but emphasizing the fragility and ephemeral nature of magnetotaxis simply supports Landler's argument.

9) Subsection “New behavioral experiments support original study”: These are reports of new studies carried out under different conditions. They don't directly address the Landler report. They should be published separately with proper peer review.

10) Subsection “Magnetic field heading over time”. "…most of the worms appeared starved and displayed a strong heading of 341° (r = 0.35, p = 0.12)." The heading is not significantly different from random. Therefore, interpretation must reflect the non-significance.

11) Subsection “Magnetic orientation in three-dimensions”: Here Vidal-Gadea reiterate their expectation that when traveling in 3D within the soil, a worm can somehow choose one specific direction among the cone of directions that have a constant angle relative to the field. This just doesn't work. In the 2D plate assay this is at least conceptually possible: the horizontal orientation of the plate leaves the worm only two such directions on the cone. And if the worm is left-right asymmetric, one can concoct a way to choose just one of those. However, in the 3D condition in soil there is no horizontal plane for reference and thus selecting one special angle among the many on the cone is not possible in principle. This argument is based on symmetry and holds regardless of how much "vector math" the worms know. If the worms were able to sense a symmetry-breaking cue, like gravity, then one needs to ask why they don't simply follow that cue.

12) Subsection “Physiological evidence for magnetoreception in *C. elegans”*, This section is redundant with the summary in the Introduction and does not address the Landler report, and so should be omitted here. The authors criticize the Keays manuscript for not discussing the AFD calcium imaging results from their 2015 paper. However, the responses in that paper to earth-strength fields were very small (1-2%∆F/F) and variable (confidence intervals for the average trace overlapping baseline), while they only showed tax-4 and synaptic transmission mutant responses to fields of 100X earth strength. Again, this is mostly a critique of the 2015 paper, but I don't think the results described in this paragraph strengthen the authors' case very much. I think they would be better off cutting this paragraph. The neural imaging results and the ecological conclusions seem much more questionable and preliminary, and should be eliminated from this manuscript.

---

## [Author Response]

Summary:The rebuttal from Vidal-Gadea has three main parts:1) A specific technical concern regarding Landler's attempt at replication. This is legitimate and Landler could address it with a reasonable bit of additional analysis.

Thank you for agreeing with our concerns that Landler et al. did not replicate experiments in our 2015 study. When trying to disprove conclusions from a study, we believe that it falls on others to first try and replicate the original experiments before modifying them. Please note that by their own description in the most recent version of their article, Landler et al. still did not replicate our assays, or our controls. We feel this is unacceptable given how our reanalysis of data presented in our Figure 2 suggests that had they run proper controls, they may have found positive results.

2) A long list of potential reasons why magnetotaxis experiments might fail. This line of argument supports Landler's conclusion that C. elegans magnetotaxis is not a robust effect and thus not a suitable model system.

Our goal here was not to provide an unsurmountable list of parameters to be met by experimenters, but rather to enumerate potential variables that may or not impact outcomes. We removed much of the items on this list which we consider to have low likelihood of affecting outcome and left only those which we know to affect this assay.

3) A report on new experiments on the subject that appear to confirm magnetotaxis in C. elegans. These don't directly address the Landler report, they use somewhat different methods, and should be published separately.

As requested, we have removed all new experiments. We only include a reanalysis of data from Landler et al., 2018 and our original 2015 study in three figures. No new data areincluded in this resubmission.

Specific comments:1) In subsection “Magnetotaxis assay”, the authors state (referring to their 2015 paper) "the cGMP-gated cation channel TAX-4 was necessary and sufficient in AFD neurons for magnetic orientation" While they did show TAX-4 was required, they did not show that is was sufficient (i.e. that expression of TAX-4 in a non-magnetically sensitive cell conferred magnetic sensitivity). In fact, TAX-4 is expressed in many sensory neurons (AWB, URX, ASE, AQR….) that are not seemingly involved in magentosensation. This statement needs to be changed.

Although we did not show or claim that *tax-4* was sufficient to convey magnetosensation in other cells, we did provide evidence that *tax-4* was sufficient in the AFD neurons for magnetic orientation. This was accomplished by showing that expression of *tax-4* only in AFD neurons, and not in any other neuron in a *tax-4* null mutant background, was sufficient to rescue magnetic orientation. We do not believe that TAX-4 alone would be sufficient for magnetic orientation. Instead, TAX-4 likely conveys the final depolarization step in a transduction pathwaylike other sensory transduction pathways operate such as in sensing odor, tastant, thermal, and gaseous cues. Nevertheless, we have removed this text to shorten our rebuttal as requested.

2) In subsection “Overt differences in experimental methods”: The main technical concern regarding Landler's experiments is that the movement assays were conducted over a longer period of time (60 min vs 30 min). Vidal-Gadea contends that over this period the worms could have changed from "fed" to "starved", leading them to switch the polarity of magnetotaxis. If so, then Landler may have observed a mix of worms moving in opposite directions. Note that this concern applies to some but not all of the Landler experiments, for example the starved worms in their Figure 3B were presumably always starved, and here Landler's results are in direct conflict with Vidal-Gadea's.

We have adjusted the text to acknowledge this.

3) In the same subsection: A long list of what look like minor differences between the experimental protocols in the two labs. For none of these differences (electrically shielded pipettes, plastic caps vs parafilm, brass vs aluminum for the sham magnet, etc.) is there a plausible argument how they would affect magnetotaxis. If in fact magnetotaxis is real, but so fragile that it depends in some intricate manner on the assay tube material or on large-scale field homogeneity or on minuscule temperature gradients, then one would have to agree with Landler that this just isn't a robust model system for magnetotaxis.

We agree that some of these factors are unlikely to influence differences in our results. To shorten our rebuttal, we have removed this discussion. We also would like to point out that magnetic orientation in *C. elegans* is not as robust a behavior as chemotaxis or touch avoidance. We have never claimed that the behavior is robust, merely that it is real and achievable by experienced researchers. Landler et al., however, argue that magnetotaxis is not real or even theoretically plausible. This is a claim that data from our labs, and the labs of others, do not support.

4) In subsection “Additional differences in experimental methods”, the authors cite the requirement for tax-4 and the AFD neurons for as evidence against magnetotaxis being an artifact of temperature gradients. But as the authors must know, AFD neurons are exquisitely sensitive to very small temperature fluctuations, and their temperature responses (which depend on tax-4) are far more robust than their responses to earth-strength magnetic fields. To this reviewer, the involvement of AFD undermines rather than supports their contention that temperature is not involved in the behavior they observe. This section should probably be cut.

We have removed this section to shorten the text as requested.

5) In subsection “Horizontal plate assay”, indeed, levels of significance reported in the original paper (Vidal-Gadea et al., 2015) were based on an inappropriate number of degrees of freedom. This is because individual worms were used as replicates although they were tested within the same Petri dish and were therefore not independent of one another. Treating them as such violates a key assumption of the statistical tests used and leads to inflated type 1 error. In the response to Lander et al. Vidal-Gadea et al. claim that reanalysis of the data using mean headings of the worms had no major effect on results. They should either refrain from this argument or present the re-analysed data (preferably in from of a supplementary table).

We addressed this issue in our rebuttal and plot all data in Figure 1. With the new way of analysis, we still find statistically significant results.

6) In subsection “Magnetotaxis assay”, it is stated that " worms in their natural habitat may have access to much greater amounts of iron than those provided for them in the lab…." We don't know what evidence the authors have for this statement. The current view on C. elegans ecology (e.g. Felix and Braendle, 2010) is that it is not a soil-dweller but rather colonizes decaying fruit and plant matter. Please remove this statement.

We adjusted the language to read that iron is abundant in soil surrounding decaying fruit and plant matter.

7) In many places the authors state that "worms migrated differently based on their global site of origin" with the specific implication that their internal compass is set to give a consistent up/down migration direction given the location of the population on the globe. These results are too preliminary to firmly draw this conclusion. Although many wild isolates of C. elegans have been described and are available from the stock center, the authors have only tested 3. Moreover, one of these, the N2 lab strain, is not really a good example of a "British" wild strain N2 was grown in liquid culture for many years (in Berkeley) followed by many more years of culture on agar plates before it was frozen. During these thousands of generations of "domestication", it is known to have acquired multiple mutations adapting it to growth in a lab environment. To establish a firm link between the local magnetic field orientation and the preferred migration direction of wild populations, it would be important to sample a wider selection of genuinely wild C. elegans isolates, and ideally cross them to get some insight into how magnetic preference segregates genetically. While this is more a critique of the original 2015 paper than anything else, it is not clear that the somewhat preliminary wild isolates data strengthens the authors' case very much.

Our finding that in six out of six conditions – 3 independent wild-type strains (N2, AB1, CB) in both fed and starved states, migrated at predicted angles that associate with the angle of inclination of their native site of isolation, and in four out of four conditions – 2 independent wild-type strains (N2 and AB1) in both fed and starved states, migrated in predicted directions in vertically aligned agar tubes supports our hypothesis. We agree that future work should address the basis for this phenomenon, but respectfully disagree that it should not be mentioned because it strongly supports our conclusions.

8) In subsection “Important experimental variables for magnetic orientation in C. elegans”: An additional long list of environmental factors that could lead magnetotaxis experiments to fail. "Because of these demanding physiological and environmental parameters that the worms must be in before they will orient to magnetic fields, we hesitate to suggest which parameter may have caused trouble in Landler et al., 2018." This raises some fundamental concerns: The claim that experiments "work" only under a very specific constellation of conditions is a feature of what Langmuir once dubbed "pathological science". This belief opens a large number of degrees of freedom that the experimenter can use for data selection. If one day the magnetotaxis assay "didn't work", the student is tempted to look at the hygrometer and conclude that the weather was either too humid or too dry and exclude that data set from analysis. Perhaps Vidal-Gadea and colleagues took special precautions to guard against such risks of self-deception, but emphasizing the fragility and ephemeral nature of magnetotaxis simply supports Landler's argument.

We overcame this concern by including all data with optimal and suboptimal conditions baring extreme ones detailed in our methods (e.g. sick worms). The wide field of behavior research is familiar with the difficulty of studying certain behaviors in lab conditions. e.g. seasonal differences in behavioral responses, effects of artificial substrate, unnatural temperature and lighting, etc.. Our goal for this rebuttal was to help other labs avoid these pitfalls by recognizing these factors. We have reduced the length of our rebuttal to describe only the factors that we feel are critical to replicate robust experiments.

9) Subsection “New behavioral experiments support original study”: These are reports of new studies carried out under different conditions. They don't directly address the Landler report. They should be published separately with proper peer review.

Without our knowledge, Carlos Caldart and Diego Golombek at the National University of Quilmes, Argentina independently reproduced our finding that *C. elegans* orients to an artificial magnet and that this orientation required cyclic-nucleotide-gated ion channel subunit TAX-2. These positive results directly address Landler et al’s concerns that our experiments cannot be reproduced. Nevertheless, we agree that they would provide stronger support if published in a separately and have thus removed them.

10) Subsection “Magnetic field heading over time”. "…most of the worms appeared starved and displayed a strong heading of 341° (r = 0.35, p = 0.12)." The heading is not significantly different from random. Therefore, interpretation must reflect the non-significance.

We have removed all new experiments as requested. Therefore, our live-tracking experiment that showed how the preferred direction of orientation changes over time from away from magnetic north, to random, and finally weakly towards magnetic north has been removed.

11) Subsection “Magnetic orientation in three-dimensions”: Here Vidal-Gadea reiterate their expectation that when traveling in 3D within the soil, a worm can somehow choose one specific direction among the cone of directions that have a constant angle relative to the field. This just doesn't work. In the 2D plate assay this is at least conceptually possible: the horizontal orientation of the plate leaves the worm only two such directions on the cone. And if the worm is left-right asymmetric, one can concoct a way to choose just one of those. However, in the 3D condition in soil there is no horizontal plane for reference and thus selecting one special angle among the many on the cone is not possible in principle. This argument is based on symmetry and holds regardless of how much "vector math" the worms know. If the worms were able to sense a symmetry-breaking cue, like gravity, then one needs to ask why they don't simply follow that cue.

We agree with the reviewer that worms need to use another cue other than the magnetic field to help break the symmetry. We have adjusted our text to focus on this hypothesis. As to why worms need to sense magnetic field if they can sense these other cues? The same question might be asked of other animals. Like other animals that orient to the magnetic field, the other cues might be unreliable. For instance, birds orient using visual cues, but rely more on magnetic cue in the absence of visual landmarks. Worms might rely on moisture, thermal or gaseous gradients to detect the vertical dimension as an allothetic cue. However, in a rotting compost pile, these gradients may not consistently reflect up versus down directions during certain conditions (e.g. rain, weather events, and decomposition). In these circumstances, the worm may rely more on magnetic cues to decide which way to move vertically.

12) Subsection “Physiological evidence for magnetoreception in C. elegans”, This section is redundant with the summary in the Introduction and does not address the Landler report, and so should be omitted here. The authors criticize the Keays manuscript for not discussing the AFD calcium imaging results from their 2015 paper. However, the responses in that paper to earth-strength fields were very small (1-2%∆F/F) and variable (confidence intervals for the average trace overlapping baseline), while they only showed tax-4 and synaptic transmission mutant responses to fields of 100X earth strength. Again, this is mostly a critique of the 2015 paper, but I don't think the results described in this paragraph strengthen the authors' case very much. I think they would be better off cutting this paragraph. The neural imaging results and the ecological conclusions seem much more questionable and preliminary, and should be eliminated from this manuscript.

From our 2015 study, average responses did not overlap with baseline and were all significantly different from baseline for 6 out of 6 experimental conditions (Figure 7C-H, K). We showed wild-type responses to 100x earth strength in Figure 7C,D,K. We also reported changes in fluorescence of 20% in our 2015 supplementary methods for partially restrained worms. Our finding that worms did not respond higher to 1x versus 100x field strength is consistent with the idea that worms have not evolved in the presence of stronger than earth magnetic fields. Nevertheless, to shorten our rebuttal as requested, we have removed this section.